# Prostaglandin analog effects on cerebrospinal fluid reabsorption via nasal mucosa

**Michelle G. Pedler**[1]**, J. Mark Petrash**[1]**, Prem S. Subramanian**[1,2,3,4]*

**1** Department of Ophthalmology, University of Colorado School of Medicine, Aurora, Colorado, United States of America, **2** Department of Neurology, University of Colorado School of Medicine, Aurora, United States of America, **3** Department of Neurosurgery, University of Colorado School of Medicine, Aurora, Colorado, United States of America, **4** Division of Ophthalmology, Uniformed Services University of the Health Sciences, Bethesda, Maryland, United States of America

* prem.subramanian@cuanschutz.edu

## Abstract

### Introduction

Cerebrospinal fluid (CSF) outflow has been demonstrated along nasal lymphatics via olfactory nerve projections; flow may be increased by stimulating lymphatic contractility using agents such as noradrenaline and the thromboxane A2 analog U46619. Lymphatics elsewhere in the body show increased contractility upon exposure to the prostaglandin F2alpha analog isoprostane-8-epi-prostaglandin. We investigated the ability of ophthalmic prostaglandin F2alpha analogs to increase CSF outflow when applied to the nasal mucosa by inhalation.

### Methods

Latanoprost (0.1, 0.5, or 1mg/ml), bimatoprost (0.3 or 3mg/ml), travoprost (0.04 or 0.4mg/ml), latanoprostene bunod (0.24 or 2.4mg/ml), tafluprost (0.25 or 2.5mg/ml), or control vehicle (10% DMSO) was administered to awake adult C57B/6 mice by nasal inhalation of 2μl droplets. Multiday dosing (daily for 3 days) of latanoprost also was evaluated. A total of 81 animals were studied including controls. General anesthesia was induced by injection, and fluorescent tracer (AlexaFluor647-labelled ovalbumin) was injected under stereotaxic guidance into the right lateral ventricle. Nasal turbinate tissue was harvested and homogenized after 1 hour for tracer detection by ELISA and fluorometric analysis.

### Results

Inhalation of latanoprost 0.5mg/ml and 1mg/ml led to a 11.5-fold increase in tracer recovery from nasal turbinate tissues compared to controls (3312 pg/ml vs 288 pg/ml, p<0.001 for 0.5mg/ml; 3355 pg/ml vs 288 pg/ml, p<0.001 for 1mg/ml), while latanoprost 0.1 mg/ml enhanced recovery 6-fold (1713 pg/ml vs 288 pg/ml, p<0.01). Tafluprost 0.25mg/ml and bimatoprost 0.3mg/ml showed a modest (1.4x, p<0.05) effect, and the remaining agents showed no significant effect on tracer recovery. After 3 days of daily latanoprost treatment and several hours after the last dose, a persistently increased recovery of tracer was found.

**Data Availability Statement:** All relevant data are within the paper and its Supporting information files.

**Funding:** This work was supported in part by an unrestricted grant to the University of Colorado

Department of Ophthalmology from Research to Prevent Blindness, Inc. The funders had no role in study design, data collection and analysis, decision to publish, or preparation of the manuscript.

**Competing interests:** The authors have declared that no competing interests exist.

## Conclusions

Prostaglandin F2alpha analogs delivered by nasal inhalation resulted in increased nasal recovery of a CSF fluorescent tracer, implying increased CSF outflow via the nasal lymphatics. The greatest effect, partially dose-dependent, was observed using latanoprost. Further studies are needed to determine the efficacy of these agents in reducing ICP in short and long-term applications.

## Introduction

Cerebrospinal fluid (CSF) is produced in the lateral ventricles by active transport across the cell membranes of epithelial cells lining the arachnoid villi and is dependent upon Na/K ion channel activity [1]. CSF then flows freely through the ventricles, fills several cisterns as well as sulci along the surface of the brain, and is maintained in homeostasis via reabsorption through several putative pathways [1], including via arachnoid granulations [2]. CSF also is present in the meningeal coverings of the cranial nerves that exist before they exit the cranial compartment [2], and CSF also can flow along the most proximal portion of spinal root ganglia and nerves [3]. Studies demonstrate that normal CSF outflow occurs along these nerves and nerve roots, although the relative importance of each pathway in the maintenance of normal CSF turnover remains controversial [1]. CSF movement along the olfactory nerves through the cribriform plate and into the nasal lymphatic system comprises a major outflow pathway in experimental animals, and it may take on greater importance when intracranial pressure (ICP) is elevated [4–7]. Because extracranial lymphatic outflow is not dependent upon venous sinus pressures, CSF flow along cranial nerves may not be affected when cerebral venous outflow is diminished, and increased CSF flow may compensate for reduced absorption via the arachnoid granulations [8].

Current treatment strategies for disorders of elevated ICP use medications to reduce CSF production by inhibition of carbonic anhydrase (acetazolamide, methazolamide), diuresis (same agents as well as furosemide), and/or sodium transport inhibition (furosemide) [9]. No medications are available that will increase CSF outflow, and the olfactory lymphatics provide an accessible target for such a treatment [10]. Although lymphatic channels are often thought to carry fluid passively into the venous system, increased lymphatic contractility can be induced pharmacologically. Experiments with lower extremity lymphatic stimulation have demonstrated improvement in peripheral edema with use of a variety of pharmacologic agents, such as the prostaglandin F2 alpha analogue isoprostane 8-epi-prostaglandin F2alpha (PGF2alpha) [11]. CSF pressure monitoring in sheep after intranasal delivery of pharmacologic agents that may affect lymphatic contractility suggests that CSF outflow facility may be manipulated in this manner [10]. PGF2alpha analogues have been used clinically for decades, and their safety and tolerability when applied to the conjunctival surface is well-studied [12]. We therefore hypothesized that commercially available PGF2alpha analogues, widely used in the treatment of open angle glaucoma, might increase CSF outflow through the nasal lymphatics in an animal model and sought to directly demonstrate this effect.

## Methods

Animal use protocols were approved by the Institutional Animal Care and Use Committee (IACUC) of the University of Colorado School of Medicine (approval #00943). Adult male

and female C57/BL6 mice (approximately 6 weeks of age) were housed at 22˚C (72˚F), 5 mice of the same sex per standard ventilated cage, with a 14 hour light/ 10 hour dark cycle. They were were provided access to water, a commercial lab animal diet, and bedding material ad libitum. Health and welfare were checked daily by a veterinary technician with the supervision of a licensed veterinarian. Animals were acclimated for 1 week before all experiments. Commercially available PGF2-alpha analogues used in the treatment of glaucoma were selected for evaluation; these included latanoprost, bimatoprost, travoprost, latanoprostene bunod, and tafluprost. Initial experiments were conducted with topical ophthalmic preparations, and pharmaceutical grade drug also was obtained and placed into solution for use in a more concentrated form. Agents were solubilized in DMSO and diluted in normal saline, with 10% DMSO serving as vehicle control.

For single inhalation trials, awake animals were administered intranasal vehicle (10% DMSO) or drug solution (total 8 μl) by placing droplets (2 μl) of liquid at the nares using a micropipette and allowing the animal to inhale the liquid spontaneously [13]. Animals were treated in a random order to limit confounding. Anesthesia with ketamine and xylazine then was administered via intraperitoneal injection, and anesthetized animals were placed within a stereotaxic apparatus. For multiday studies, intranasal drug administration was performed on awake animals in the same fashion daily, and animals were monitored per standard protocol by the veterinary technician; approximately 6 hours after the third daily dose was given, animals were anesthetized and positioned as above. A midline incision was created along the scalp, and the right lateral ventricle was targeted with placement of a 26 ga needle to a depth of 3 mm into the brain at a position 1 mm lateral and 0.3 mm anterior to bregma [14]. A 5 μl intraventricular injection of fluorescent tracer (AlexaFluor[647]-ovalbumin solution) was delivered at a rate of 100 nl/sec using a Hamilton syringe and an electric pump. Animals remained anesthetized for the 1 hour recovery period post injection and were then euthanized with a lethal overdose of ketamine followed by secondary cervical dislocation. Nasal turbinates were harvested, and tissue extracts were analyzed by ELISA in a masked fashion.

The statistical plan was developed with an expectation that tracer recovery would act as a linear variable. Sample size calculations were not informative, and group sizes were chosen empirically. No data points were excluded. Statistical analyses were conducted with Stata (College Station, TX) for two-tailed Student's t-tests, and ANOVA RM with Dunnet's post hoc analysis were performed with Graphpad Prism (Graphpad Software Inc, La Jolla, CA, USA) for comparisons between groups. All asterisks are compared to 10% DMSO vehicle control unless otherwise indicated with a line bar. Asterisks refer to p values where *p<0.05, **p<0.01 or ***p<0.001.

## Results

The experimental technique was validated by assessing anatomic placement of the tracer into the lateral ventricle (Fig 1) and by recovery of fluorescent tracer from nasal mucosa after ventricular injection in the absence of intranasal treatment. Timepoints for tissue harvest were selected based on increased recovery at 60 min versus 30 min, with later harvest raising concern for potential contamination of specimens by hematogenous CSF absorption of tracer and recirculation throughout the tissues via arterial flow. Other tissues (spleen, liver) from injected animals did not demonstrate significant levels of ovalbumin tracer relative to uninjected animals after 60 minutes.

A total of 67 animals were studied using a scaled dosing technique in which each prostaglandin analog was evaluated at its commercial ophthalmic concentration as well as one or two more concentrated levels (either 5x or 10x). When inhaled in their commercial

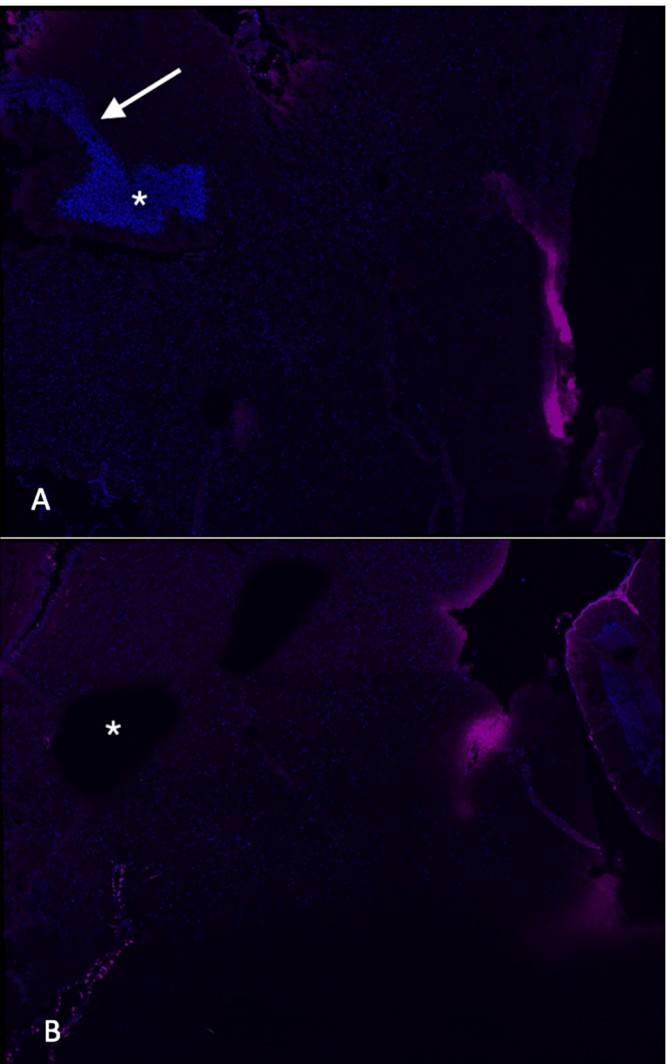

**Fig 1. Evaluation of fluorescent tracer injection.** A. Photomicrograph (20X magnification) of brain harvested after stereotaxic injection of labelled ovalbumin into the right lateral ventricle showing localized signal within the ventricle (asterisk) and along the injection track (arrow). B. Section from a control, uninjected animal demonstrating no staining in the ventricle (asterisk).

ophthalmic formulations, both bimatoprost and tafluprost showed an approximately 1.5 fold increase in nasal tracer recovery relative to vehicle-treated controls (Fig 2, p<0.05). Neither drug showed a significant effect when used at a higher concentration; additionally, neither travoprost nor latanoprostene bunod demonstrated the ability to increase nasal tracer recovery above control levels at any concentration used (Fig 2). Intranasal delivery of latanoprost resulted in a semi dose-dependent increase in tracer recovery from the nasal mucosa after 60 min, with approximately 10-fold increased levels observed relative to control animals when 0.5 mg/ml or 1 mg/ml solution was used (Fig 3, p<0.001 compared to control). The standard ophthalmic dose, 0.1 mg/ml, resulted in approximately 6-fold increased tracer recovery (Fig 3, p<0.01 vs control).

An additional 14 animals were studied several hours after receiving the third of 3 daily inhaled doses of latanoprost (either 0.1, 0.5, or 1 mg/ml). A nearly 3-fold increase in tracer

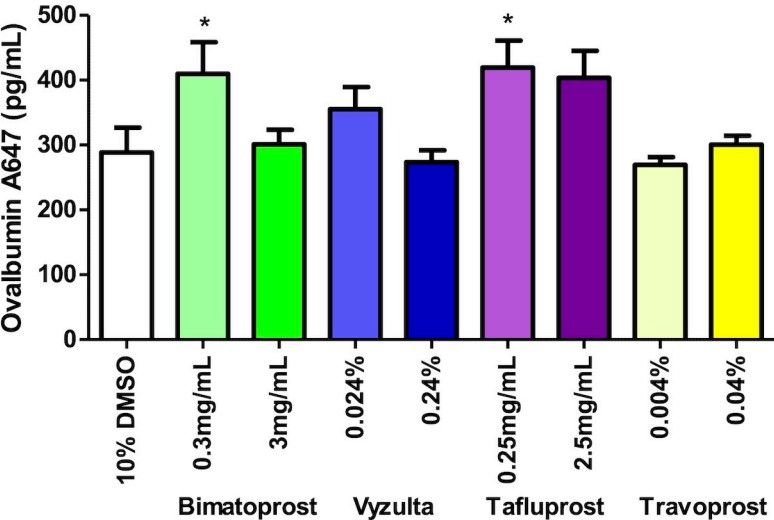

**Fig 2. Fluorometric analysis of nasal turbinate homogenates after prostaglandin F2alpha agonist bimatoprost, latanoprostene bunod, tafluprost, and travoprost inhalation and ventricular tracer injection.** Slightly enhanced tracer recovery relative to controls was seen with the lower doses of bimatoprost and tafluprost. Error bars denote SEM. Asterisks indicate $p < 0.05$ relative to controls.

recovery relative to controls ($p < 0.05$) was observed, with no statistically significant difference observed amongst the 3 doses (Fig 4).

## Discussion

In this study, latanoprost showed a dose-dependent ability to increase recovery of a CSF-based fluorescent tracer substance from the nasal mucosa of experimental animals shortly after intranasal inhalation. A lesser effect was observed with tafluprost and bimatoprost, and these drugs did not appear to have a dose-dependent effect. Two additional prostaglandin analogues, latanoprostene bunod and travoprost, showed no apparent impact on tracer recovery when

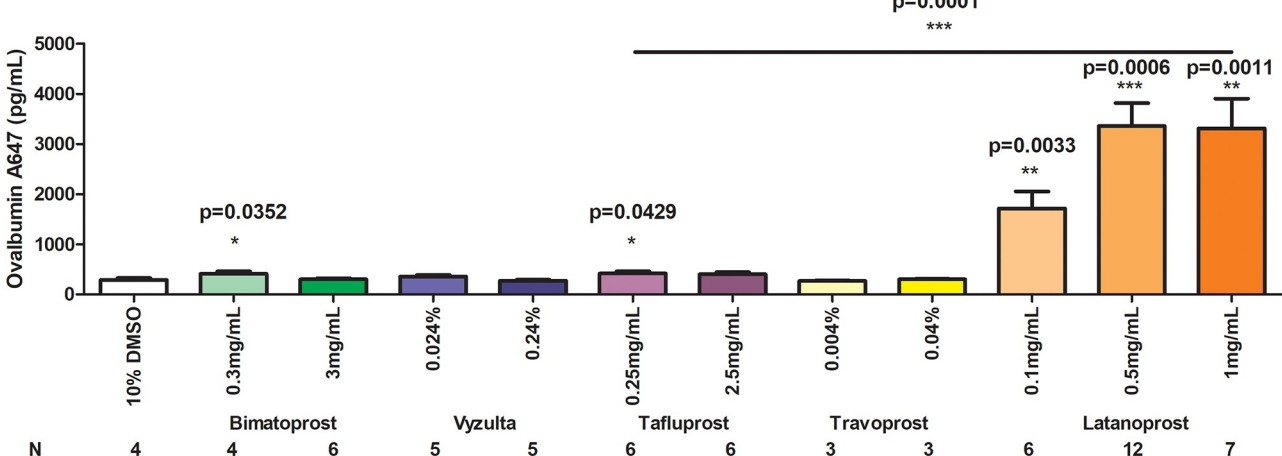

**Fig 3. Comparison of the effect of latanoprost intranasal application to other PGF2alpha analogues in the recovery from nasal turbinates of fluorescently tagged ovalbumin after CSF injection after single dose application of each drug.** All statistical comparisons are relative to controls except as indicated by the horizontal bar comparing 1 mg/ml latanoprost with 0.25 mg/ml tafluprost; * = $p < 0.05$, ** = $p < 0.01$, *** = $p < 0.001$.

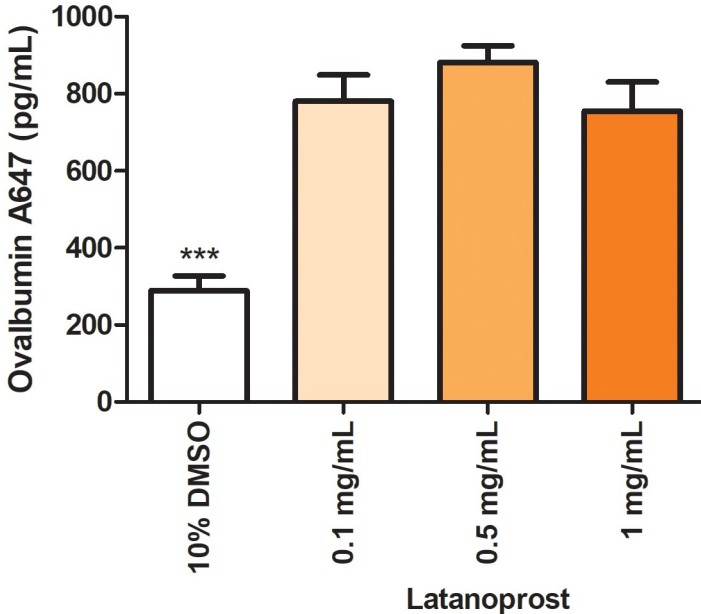

**Fig 4. Evaluation of tracer recovery from the nasal turbinates of animals treated with daily doses of latanoprost over 3 days (4 animals treated with 0.1 mg/ml; 5 each with the other two doses).** Nasal turbinates were isolated 1 hour after ventricular tracer injection and between 7 and 9 hours after the final latanoprost dose was given intranasally; * = p<0.01 relative to control.

inhaled intranasally. Latanoprost, when given in daily doses for 3 days, showed a sustained ability to increase tracer recovery even several hours after the last dose was administered.

Prior work with pharmacologic agents introduced into the nasal cavities of sheep by nebulization showed an increase in CSF outflow as measured by an intracranial pressure monitoring system by which fluid was infused into the lateral ventricle and the resulting pressure changes assessed [10]. In this system, it was calculated that a 2.29-fold to 2.44 fold increase in CSF outflow was achieved with the use of either NG-monomethyl L-arginine or U46619, a thromboxane A2 analogue. Interestingly, while the latter agent increases lymphatic contractility, the former inhibits nitric oxide synthesis and causes smooth muscle relaxation, and the authors postulated that an increased calibre of the lymphatic channels might allow greater passive flow along a pressure gradient in some instances [10].

The relative contribution of nasal lymphatic drainage to overall CSF outflow has not been determined in humans. In both sheep and rat models, injection of a radiolabelled tracer into the CSF with recovery from lymphatic and other sources suggests that at least 50% of normal CSF drainage in awake animals may occur via the lymphatic system [15, 16]. Additionally, induced communicating hydrocephalus in a rodent model is associated with reduced CSF lymphatic outflow along the olfactory and nasal mucosal pathways [6].

We observed that latanoprost had a stronger effect on CSF tracer recovery than did other prostaglandin F2alpha analogues. Prior studies have shown a variable effect of different pharmacologic agents of the same class on vascular contractility. The isoprostane 8-epi-PGF2alpha has a more potent effect on lymphatic channels than does U46619 [11], and application of U46619, PGF2alpha itself, and latanoprost had a significantly greater effect on porcine ciliary artery constriction than did travoprost [17]. Latanoprost also causes increased lymphatic outflow from the eye in mice [18], and a diminished effect of latanoprost on intraocular pressure has been observed in humans who have undergone surgical ligation of cervical lymphatics for

cancer treatment [19]. However, the relative effect of other PGF2alpha analogues on ocular lymphatic drainage has not been reported.

In conclusion, we found that a single nasal inhalation of the PGF2alpha analogue latanoprost resulted in an increased recovery of a CSF tracer from nasal mucosa, indicating greater passage of CSF through olfactory lymphatic channels, with a weaker effect seen with bimatoprost and tafluprost. We found multiday dosing also had a sustained effect even several hours after nasal inhalation of drug. Further study is needed to assess the local effect on nasal mucosa from acute and chronic application of latanoprost solutions and to determine its tolerability in humans, although experience with conjunctival application would indicate low risk of adverse events [12]. Based on ocular studies [18], we anticipate that lymphatic activity induced by latanoprost would persist and not diminish over time. If the drug can be used intranasally on a chronic basis, it may be a useful therapeutic agent for humans with disorders of ICP elevation.

## Supporting information

**S1 Data.**
(XLSX)

**S2 Data.**
(XLSX)

**S3 Data.**
(XLSX)

**S4 Data.**
(XLSX)

**S5 Data.**
(XLSX)

## Author Contributions

**Conceptualization:** J. Mark Petrash, Prem S. Subramanian.

**Formal analysis:** Michelle G. Pedler.

**Investigation:** Michelle G. Pedler, Prem S. Subramanian.

**Methodology:** Michelle G. Pedler, J. Mark Petrash, Prem S. Subramanian.

**Resources:** J. Mark Petrash, Prem S. Subramanian.

**Supervision:** Prem S. Subramanian.

**Writing – original draft:** Prem S. Subramanian.

**Writing – review & editing:** Michelle G. Pedler, J. Mark Petrash, Prem S. Subramanian.

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
