## [Decision Letter · Decision Letter 0]

17 Jun 2021

PONE-D-21-06439

Prostaglandin analog effects on cerebrospinal fluid reabsorption via nasal mucosa

PLOS ONE

Dear Dr. Subramanian,

Thank you for submitting your manuscript to PLOS ONE. After careful consideration, we feel that it has merit but does not fully meet PLOS ONE’s publication criteria as it currently stands. Therefore, we invite you to submit a revised version of the manuscript that addresses the points raised during the review process.

We look forward to receiving your revised manuscript.

Kind regards,

Francesco Lolli, M.D., Ph.D.

Academic Editor

PLOS ONE

Additional Editor Comments:

The manuscript is technically sound, and the data do support the conclusions. However both reviewer had major comments, and some involve additional data requested

Journal Requirements:

2. PLOS ONE has specific criteria regarding the reporting of animal research (https://journals.plos.org/plosone/s/submission-guidelines#loc-animal-research). Specifically, these guidelines require that details regarding care, monitoring, and method of sacrifice are clearly stated. To that effect, please include the following in your Methods section:

- A description of the animals' housing conditions, including food/water, temperature, and lighting conditions

- How animals were monitored for health and behavior throughout the study, and the timeline of the study protocol

As part of your revision, please complete and submit a copy of the ARRIVE Guidelines checklist, a document that aims to improve experimental reporting and reproducibility of animal studies for purposes of post-publication data analysis and reproducibility: https://arriveguidelines.org/sites/arrive/files/Author%20Checklist%20-%20Full.pdf. Please include your completed checklist as a Supporting Information file. Note that if your paper is accepted for publication, this checklist will be published as part of your article.

Thank you for your attention. We look forward to hearing from you.

Reviewers' comments:

Reviewer's Responses to Questions

**Comments to the Author**

1. Is the manuscript technically sound, and do the data support the conclusions?

Reviewer #1: Yes

Reviewer #2: Yes

2. Has the statistical analysis been performed appropriately and rigorously? 

Reviewer #1: Yes

Reviewer #2: Yes

3. Have the authors made all data underlying the findings in their manuscript fully available?

Reviewer #1: Yes

Reviewer #2: Yes

4. Is the manuscript presented in an intelligible fashion and written in standard English?

Reviewer #1: Yes

Reviewer #2: Yes

5. Review Comments to the Author

Reviewer #1: In this article, nasal delivery of prostaglandin analogues boosted nasal recovery of a CSF fluorescent tracer, implying enhanced CSF outflow via the nasal lymphatics. Latanoprost had the most substantial impact and dose-dependent. These medicines may help lower ICP.

Some section of the article needs a better presentation.

The statistic section describes "Student's t-tests and ANOVA RM with Dunnet's post hoc analysis". It is not informative and should be expanded, referring to preplanned methods.

The results refer to Fig.2 and Fig. 3, but it is not clear how the asterisk refers to the difference between the groups in which specific test.

Fig legends refer more to methods than results.

Fig. 1a and 1b should show the area of interest. Please indicate the signals within the ventricle and the injection track (Fig.1A) and the control ventricles with no staining (Fig.1B).

Reviewer #2: The authors investigated the ability of ophthalmic prostaglandin F2alpha analogs to increase CSF outflow when applied to the nasal mucosa by inhalation. The idea behind this work is that no medications are available that will increase CSF outflow, and the olfactory lymphatics provide an accessible target for such a treatment. The authors utilized commercially available PGF2alpha analogues (latanoprost, bimatoprost, travoprost, latanoprostene bunod, and tafluprost) that are widely used in the treatment of open angle glaucoma.

The main results is that a single nasal inhalation of the PGF2alpha analogue latanoprost resulted in an increased recovery of a CSF tracer from nasal mucosa, that suggested an increased passage of CSF through olfactory lymphatic channels, a little effect was seen with bimatoprost and tafluprost.

The work is technically well done and well executed and data obtained are very promising. The identification of compounds with the potential to increase CSF outflow is of great interest. However, it is surprising that the authors fell short of performing additional experiments to validate their important observation. Indeed, time course experiments to pinpoint the duration of a single (or even more important) repeated dose regimen are missing. Also, in my opinion it is vital to address the effects of these prostaglandin analogs on the nasal mucosa such as morphology and cell infiltrate.

In summary, this work is a nice piece of science portraying extremely interesting preliminary results but that requires additional work to be accepted for publication

6. PLOS authors have the option to publish the peer review history of their article (what does this mean?). If published, this will include your full peer review and any attached files.

Reviewer #1: No

Reviewer #2: No

---

## [Author Response · Author response to Decision Letter 0]

26 Sep 2021

Dear Dr. Lolli:

We thank you and the reviewers for your comments and suggestions to improve our manuscript. We have made the following changes and offer our responses to these comments below:

Reviewer 1

In this article, nasal delivery of prostaglandin analogues boosted nasal recovery of a CSF fluorescent tracer, implying enhanced CSF outflow via the nasal lymphatics. Latanoprost had the most substantial impact and dose-dependent. These medicines may help lower ICP.

RESPONSE: Thank you for your comments and your review.

Some section of the article needs a better presentation.

The statistic section describes "Student's t-tests and ANOVA RM with Dunnet's post hoc analysis". It is not informative and should be expanded, referring to preplanned methods.

RESPONSE: We have revised the final paragraph of the Methods section (lines 109-114 of the tracked changes version) to provide additional statistical information as requested.

The results refer to Fig.2 and Fig. 3, but it is not clear how the asterisk refers to the difference between the groups in which specific test.

RESPONSE: We have provided clarification regarding the meaning of the asterisks in the Figure legends; comparisons in each instance are between the specific drug/dose and controls unless otherwise indicated.

Fig legends refer more to methods than results.

RESPONSE: Figure legends have been rewritten to more appropriately reflect and describe the results and no longer contain methodologic information. 

Fig. 1a and 1b should show the area of interest. Please indicate the signals within the ventricle and the injection track (Fig.1A) and the control ventricles with no staining (Fig.1B).

RESPONSE: Asterisks and arrows have been added to direct the reader’s attention to the fluorescent signal in the ventricle and the injection track, respectively. 

Reviewer 2

The authors investigated the ability of ophthalmic prostaglandin F2alpha analogs to increase CSF outflow when vapplied to the nasal mucosa by inhalation. The idea behind this work is that no medications are available that will increase CSF outflow, and the olfactory lymphatics provide an accessible target for such a treatment. The authors utilized commercially available PGF2alpha analogues (latanoprost, bimatoprost, travoprost, latanoprostene bunod, and tafluprost) that are widely used in the treatment of open angle glaucoma. 

The main results is that a single nasal inhalation of the PGF2alpha analogue latanoprost resulted in an increased recovery of a CSF tracer from nasal mucosa, that suggested an increased passage of CSF through olfactory lymphatic channels, a little effect was seen with bimatoprost and tafluprost.

RESPONSE: Thank you for your review and your very helpful comments. 

The work is technically well done and well executed and data obtained are very promising. The identification of compounds with the potential to increase CSF outflow is of great interest. However, it is surprising that the authors fell short of performing additional experiments to validate their important observation. Indeed, time course experiments to pinpoint the duration of a single (or even more important) repeated dose regimen are missing. 

RESPONSE: We appreciate this recommendation to look at duration of effect of dosing. As you noted, multiple dose/multiday dosing would reflect real-life use of a medication. Therefore, we undertook additional experiments with latanoprost only (given that it showed the greatest effect with single dosing) and found a sustained effect upon CSF drainage via the lymphatic pathways (new Figure 4 and text lines 164-181). 

Also, in my opinion it is vital to address the effects of these prostaglandin analogs on the nasal mucosa such as morphology and cell infiltrate.

RESPONSE: The effect upon latanoprost and other prostaglandin analogues on the ocular/conjunctival mucosa has been studied extensively during pre-clinical and clinical studies. No concerning findings with respect to inflammation or cellular infiltrates were identified. These drugs are known to cause slight conjunctival hyperemia and other changes to goblet cells that are of no functional consequence. With limited research resources and a need to prioritize the most salient experiments because of COVID-19 pandemic-related restrictions on laboratory usage and capacity, we respectfully submit that the existing data on mucosal effects of prostaglandin analogues should be sufficient to allay any concerns regarding safety and tolerability when applied to the nasal mucosa, a tissue with similar properties. We have added a statement about the known effects of these substances on the mucosal surface (Lines 72-73) and included a new reference (12). 

In summary, this work is a nice piece of science portraying extremely interesting preliminary results but that requires additional work to be accepted for publication

RESPONSE: We thank the reviewer again for the very helpful suggestions to improve the impact of our work.

Additional Editor Comments

The manuscript is technically sound, and the data do support the conclusions. However both reviewer had major comments, and some involve additional data requested

RESPONSE: We believe we have addressed the requests for changes and additional data that the reviewers have helpfully suggested. 

Sincerely yours,

Prem S. Subramanian, MD, PhD

Professor of Ophthalmology, Neurology, and Neurosurgery

Vice Chair for Academic Affairs

Division Head, Neuro-Ophthalmology

---

## [Editor Report · Decision Letter 1]

14 Oct 2021

PONE-D-21-06439R1Prostaglandin analog effects on cerebrospinal fluid reabsorption via nasal mucosaPLOS ONE

Dear Dr. Subramanian,

Thank you for submitting your manuscript to PLOS ONE. After careful consideration, we feel that it has merit but does not fully meet PLOS ONE’s publication criteria as it currently stands. Therefore, we invite you to submit a revised version of the manuscript that addresses the points raised during the review process.

Reviewing the new versions of the  manuscript entitled “Prostaglandin analog effects on cerebrospinal fluid reabsorption via nasal mucosa”  the points raised by the reviewers were considered and responded to. The manuscript is accordingly improved.

In the present revision we noticed that certain details of animal experimental techniques, particularly a description of care/monitoring information, are absent or insufficiently described from the methods.They should be introduced.

The requests for animal reasearch are found in https://journals.plos.org/plosone/s/submission-guidelines#loc-animal-research  

 Please submit your revised manuscript by Nov 28 2021 11:59PM. If you will need more time than this to complete your revisions, please reply to this message or contact the journal office at plosone@plos.org. Please include the following items when submitting your revised manuscript:

We look forward to receiving your revised manuscript.

Kind regards,

Francesco Lolli, M.D., Ph.D.

Academic Editor

PLOS ONE

Journal Requirements:

Additional Editor Comments (if provided):

Reviewing the new versions of the manuscript entitled “Prostaglandin analog effects on cerebrospinal fluid reabsorption via nasal mucosa” the points raised by the reviewers were considered and responded to. The manuscript is accordingly improved.

In the present revision we noticed that certain details of animal experimental techniques, particularly a description of care/monitoring information, are absent or insufficiently described from the methods.They should be introduced.

The requests for animal reasearch are found in https://journals.plos.org/plosone/s/submission-guidelines#loc-animal-research
---

## [Author Response · Author response to Decision Letter 1]

2 Nov 2021

Francesco Lolli, MD, PhD

Academic Editor

PLoS ONE

Dear Dr. Lolli:

We thank you and the reviewers for your additional comments and suggestions to bring the manuscript within the PLoS ONE guidelines. We have made the following changes and offer our responses to these comments below:

Comment: Reviewing the new versions of the manuscript entitled “Prostaglandin analog effects on cerebrospinal fluid reabsorption via nasal mucosa” the points raised by the reviewers were considered and responded to. The manuscript is accordingly improved.

RESPONSE: Thank you very much.

In the present revision we noticed that certain details of animal experimental techniques, particularly a description of care/monitoring information, are absent or insufficiently described from the methods. They should be introduced.

RESPONSE: We have added more specific information describing the care and monitoring of the animals and trust that these details will fulfill the requirements (lines 69-74 and 91-92 in the tracked changes version). We used a recent article in PLoS ONE (https://doi.org/10.1371/journal.pone.0247149) as a model after reviewing overall requirements. 

Thank you again for consideration of our work.

Sincerely yours,

Prem S. Subramanian, MD, PhD

Professor of Ophthalmology, Neurology, and Neurosurgery

Vice Chair for Academic Affairs

Division Head, Neuro-Ophthalmology

---

## [Editor Report · Decision Letter 2]

8 Nov 2021

Prostaglandin analog effects on cerebrospinal fluid reabsorption via nasal mucosa

PONE-D-21-06439R2

Dear Dr. Subramanian,

We’re pleased to inform you that your manuscript has been judged scientifically suitable for publication and will be formally accepted for publication once it meets all outstanding technical requirements.

Kind regards,

Francesco Lolli, M.D., Ph.D.

Academic Editor

PLOS ONE

Additional Editor Comments (optional):

The authors added more specific information describing the animals' care and monitoring. They did meet the requirements.
---

## [Editor Report · Acceptance letter]

23 Dec 2021

PONE-D-21-06439R2 

Prostaglandin analog effects on cerebrospinal fluid reabsorption via nasal mucosa 

Dear Dr. Subramanian:

I'm pleased to inform you that your manuscript has been deemed suitable for publication in PLOS ONE. Congratulations! Your manuscript is now with our production department. 

Kind regards, 

on behalf of

Dr. Francesco Lolli 

Academic Editor

PLOS ONE